# Tailoring the evolution of BL21(DE3) uncovers a key role for RNA stability in gene expression toxicity

Sophia A. H. Heyde[1] & Morten H. H. Nørholm [1 ✉]

Gene expression toxicity is an important biological phenomenon and a major bottleneck in biotechnology. *Escherichia coli* BL21(DE3) is the most popular choice for recombinant protein production, and various derivatives have been evolved or engineered to facilitate improved yield and tolerance to toxic genes. However, previous efforts to evolve BL21, such as the Walker strains C41 and C43, resulted only in decreased expression strength of the T7 system. This reveals little about the mechanisms at play and constitutes only marginal progress towards a generally higher producing cell factory. Here, we restrict the solution space for BL21(DE3) to evolve tolerance and isolate a mutant strain Evo21(DE3) with a truncation in the essential RNase E. This suggests that RNA stability plays a central role in gene expression toxicity. The evolved *rne* truncation is similar to a mutation previously engineered into the commercially available BL21Star(DE3), which challenges the existing assumption that this strain is unsuitable for expressing toxic proteins. We isolated another dominant mutation in a presumed substrate binding site of RNase E that improves protein production further when provided as an auxiliary plasmid. This makes it easy to improve other BL21 variants and points to RNases as prime targets for cell factory optimisation.

[1] Novo Nordisk Foundation Center for Biosustainability, Technical University of Denmark, Lyngby, Denmark. ✉email: morno@biosustain.dtu.dk

Recombinant production of proteins enables biochemical and structural studies, and the pET vectors hosted in *Escherichia coli* strain BL21(DE3) are the most popular approach in research laboratories[1]. Fast growth, high cell density, inexpensive culturing, availability of more than 100 pET expression vectors[2] combined with the detailed knowledge of *E. coli's* genetics, physiology and metabolism make it the preferred laboratory workhorse.[3] This is evident from the more than 200,000 research publications that have cited the use of the pET vectors, and that currently more than 128,000 (86% of the recombinant) structures in the Protein Data Bank and approximately half of the proteins produced worldwide for research and commercial use are produced in *E. coli*[2,4]. This remarkable success is despite that bacteria often show impaired growth and fitness-loss when being used for protein production[5]—a problem that is highly gene-specific, and we still lack clear guiding principles for gene and cell factory optimisation[6].

Membrane proteins (MPs) are important drug targets and serve essential roles in basic cellular mechanisms. In both prokaryotes and eukaryotes, 20–30% of all genes encode membrane proteins.[7] MPs are involved in fundamental mechanisms such as transport of nutrients and signal molecules, response to environmental changes, membrane stability, maintenance of the redox potential, defence and energy conversion. With natural abundances often too low to isolate sufficient material for in vitro studies, structural and biochemical investigations are limited by our ability to produce and purify MPs recombinantly in a functional state.

MPs are also notoriously known for causing burden in expression systems. They need to be properly inserted into the membrane and cannot be produced in inclusion bodies. This often leads to an overload of the membrane translocation machinery and has previously been reported to severely hamper protein homoeostasis in the cytoplasm leading to the accumulation of aggregates of proteases, chaperones, and overexpressed MPs[5,8,9]. In addition, competition for the limited space of the membrane causes decreased levels of key respiratory chain complexes and upregulation of the Arc two-component system, indicating serious alterations in central metabolism[5].

Experimental evolution has shown great capacity to provide fundamental insights into non-intuitive molecular mechanisms that escape discovery by hypothesis-driven research[10]. One solution to the gene expression burden is trivial: any mutation that will decrease the expression of the recombinant gene will provide a selective advantage. Previous attempts to evolve BL21 (DE3) for better protein production have repeatedly demonstrated this. DE3 strains carry the T7 RNA polymerase (RNAP) gene under the control of the *lacUV5* promoter, a stronger version of the native *lacZ* promoter inducible with IPTG. This allows the expression of any gene of interest under the control of a T7 promoter. Different strains have been isolated in which the toxicity of membrane protein production is reduced, leading to improved production yields: the Walker strains C41(DE3) and C43(DE3)[11], evolved in the late 1990s to tolerate overproduction of membrane proteins, the more recently characterised derivative mutant56[12] evolved for higher production of the toxic membrane protein YidC, and the strains C44(DE3) and C45(DE3)[13] evolved similarly. In all these cases, gene expression induced by IPTG inhibited colony formation on agar plates before mutations occurred and tolerance was achieved mainly due to reduced T7 RNAP activity, either via *lacI* mutations[14], or via promoter modifications, point mutations, or truncations in the T7 RNAP gene.

Here, we aimed at learning more about the gene expression burden, hoping to isolate new types of mutations in the genome of BL21(DE3) by restricting the evolutionary solution space for tolerance on three different levels: we deleted the genomic homology region known to frequently recombine with the λDE3 lysogen in BL21(DE3), we coupled expression of the toxic gene to expression of both green fluorescent protein (GFP) and an antibiotic resistance gene, and we allowed bacterial colonies to form prior to induction of toxic gene expression and isolated mutants over a week-long incubation period.

## Results

### Restricting the evolutionary space of BL21(DE3) for the production of toxic proteins.
In DE3 strains, the strong *lacUV5* promoter drives expression of T7 RNAP, and it was previously shown that homologous recombination between *lacUV5* and the weaker wild-type *lac* promoter is a dominating event that within hours of introducing a toxic gene leads to tolerance in BL21 (DE3)[15]. To prevent this, we first genetically restricted BL21 (DE3), creating a Δ*lacI*Δ*lacZ* variant by deleting exactly the part of the native *lac* locus that shared homology with the λDE3 locus (Fig. 1). In addition, this should prevent the expression of *lacY*, encoding the lactose permease, allowing a more uniform, concentration-dependent entry of IPTG into all cells in bacterial populations[16].

Next, similar to a previous study[12], we chose the *E. coli* membrane protein insertase YidC coupled to GFP to serve as a model protein to investigate stress caused by membrane protein overproduction as it formerly was shown to have a strong negative fitness effect on *E. coli*[17,18]. The GFP fusion comes in handy for phenotypic restriction because it allows visual screening for mutants that still produce the fusion protein. We further restricted the evolutionary solution space by introducing a hairpin structure in the expression plasmid that couples YidC-GFP to the expression of a ß-lactamase gene[19], providing resistance to ampicillin (Fig. 1). This way, the formation of non-producing populations should be minimised in the presence of the antibiotic.

Finally, in contrast to the previous studies[11,13,20], we aimed at introducing the protein production stress after bacterial colonies had established on plates. This spatiotemporally different approach should allow the formation of a large number of bacteria in a state of dormancy and in a structured environment, which previously was shown to constitute a unique evolutionary environment[21]. Furthermore, we speculated that mutants would be easy to identify as fluorescent secondary colonies, so-called papillae, outgrowing from the initially established colonies. To this end, two-layered agar plates were poured, allowing slow IPTG and ampicillin diffusion from the bottom to the top layer to grant sufficient time for colony formation before YidC-GFP production was induced (Fig. 1).

During incubation for 1 week at 37 °C, we observed several fluorescent papillae (Fig. 1) that were restreaked to confirm the fluorescent and ampicillin-resistant phenotype. Based on the fluorescence phenotype, the most promising mutants were isolated and cured of the YidC-GFP plasmid using a mild and simple CRISPR-based approach to plasmid curing[22]. These strains were subsequently retransformed with the original YidC-GFP plasmid to ensure that mutations leading to tolerance were localised on the genome and not on the expression plasmid. A single clone (Evo21(DE3)) was chosen for further characterisation.

### Characterisation of Evo21(DE3)'s protein production capabilities.
To benchmark Evo21(DE3), we compared its ability to produce the YidC-GFP-fusion protein to the non-evolved BL21 (DE3) wild-type strain as well as the derivative Mt56(DE3) previously described to be optimised for YidC-GFP overproduction[12].

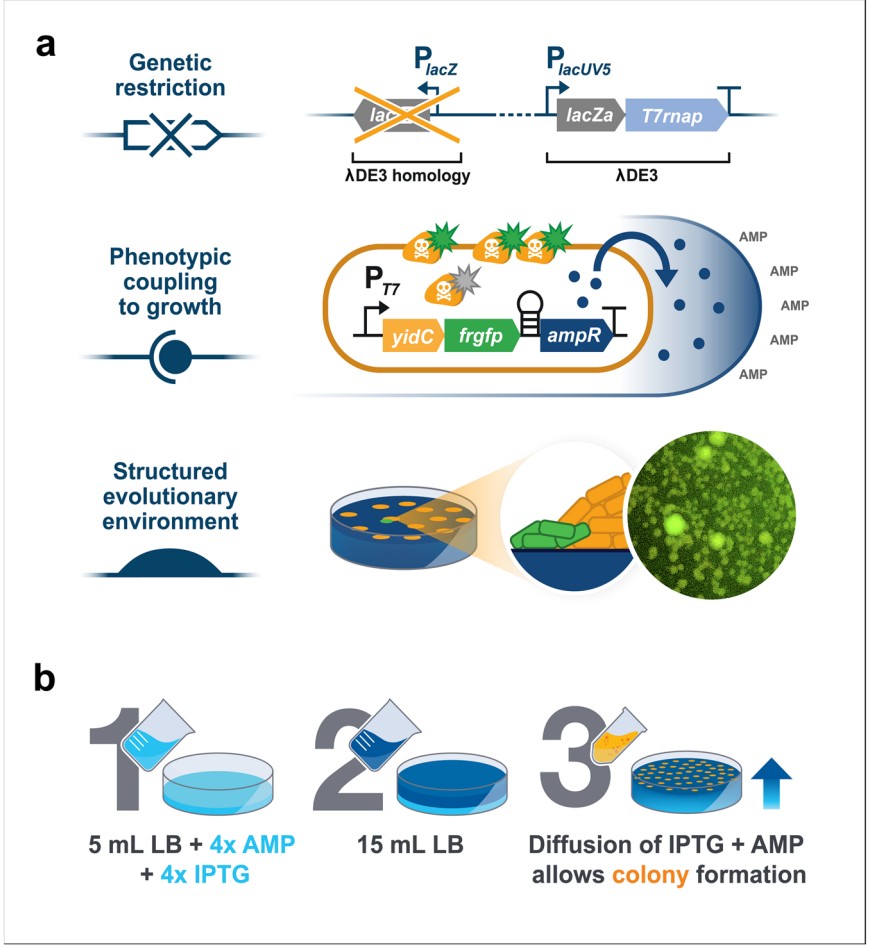

**Fig. 1 Evolution strategy to isolate mutations conferring tolerance towards toxic proteins in *E. coli* BL21(DE3). a** Illustration showing the experimental set-up used to restrict the evolutionary space for BL21(DE3) to overcome protein production toxicity on three levels: (upper panel) genetic restriction was accomplished by deleting a part of the BL21(DE3) genome that frequently recombines and lowers the T7 RNAP expression, (middle panel) protein production was coupled to both fluorescence and antibiotic resistance to prevent the formation of non-producing mutants and (lower panel) a spatiotemporally structured environment of ageing bacterial colonies was designed in order to identify strong phenotypic mutants. **b** Schematic of the workflow to produce layered agar plates that allow diffusion of the antibiotic ampicillin (AMP) and the inducer IPTG to allow sufficient time for colony formation on the surface of the top agar layer before toxic protein production is induced.

The plasmid pLysS was utilised to limit basal T7 RNAP expression[23] in BL21(DE3) and Evo21(DE3). Co-expression of pLysS in Mt56(DE3) is not relevant due to the greatly reduced polymerase activity in this strain. In liquid culture, based on GFP fluorescence, Evo21(DE3) expressed significantly ($P \leq 0.0001$) higher amounts of YidC-GFP than BL21(DE3) and Mt56(DE3). Eight hours post induction, yields were 3.6-fold higher in cultures of Evo21(DE3) than in cultures of BL21(DE3) and 2.1-fold higher than in Mt56(DE3) (Fig. 2a). Parallel monitoring of the optical density of the cultures showed no major growth impairment for the strains (Supplementary Fig. 1).

To assess whether the Evo21(DE3) phenotype was gene-specific, we next investigated the expression of a set of 24 different GFP-fusion proteins selected from an expression vector library of the *E. coli* inner membrane proteome[24]. These membrane proteins were selected to cover a wide range of functions, toxicity (previously reported as a change in $OD_{600}$ upon IPTG addition[24]), and the number of predicted transmembrane domains (Supplementary Table 1). Comparing the fold change of protein production in Evo21(DE3) and BL21(DE3), titre was improved for 19 of the 24 proteins—with a significant fold change of more than 1.5-fold for 14 of them (Fig. 2b). The highest improvement was 6.1-fold ($P \leq 0.001$), observed for the protein YihG.

As a first test that the underlying mechanism allowing Evo21 (DE3) to produce more toxic protein was not related to a general decrease in the activity of the T7 expression system—as previously observed for the BL21(DE3) derivatives C41/43 (DE3), C44/45(DE3) and Mt56(DE3)—we Sanger sequenced the T7 RNAP gene, which confirmed an absence of mutations. Next, we compared the expression of two non-toxic soluble proteins, GFP and a camelid-derived nanobody, and both were produced at higher levels in Evo21(DE3) than in BL21(DE3) or Mt56(DE3)—both in the presence and absence of pLysS (Fig. 2c, d). Similarly, Evo21(DE3) outperformed other strains in the production of seven out of eight different plant-derived cytochrome P450 enzymes—a class of enzymes of significant biotechnological interest[25] (Supplementary Fig. 2). This indicates that the causative mutation in Evo21(DE3) is different from previously isolated BL21(DE3) derivatives and probably does not cause a general decrease in T7 RNAP activity.

Even though the screening for improved protein productivity was performed with the highly efficient T7 system, the ideal mutant strain would be capable of producing more protein independently from the promoter system. To explore if this was the case for Evo21(DE3), we replaced the T7 promoter with the L-rhamnose inducible *rhaBAD* promoter in the *yidC-GFP*

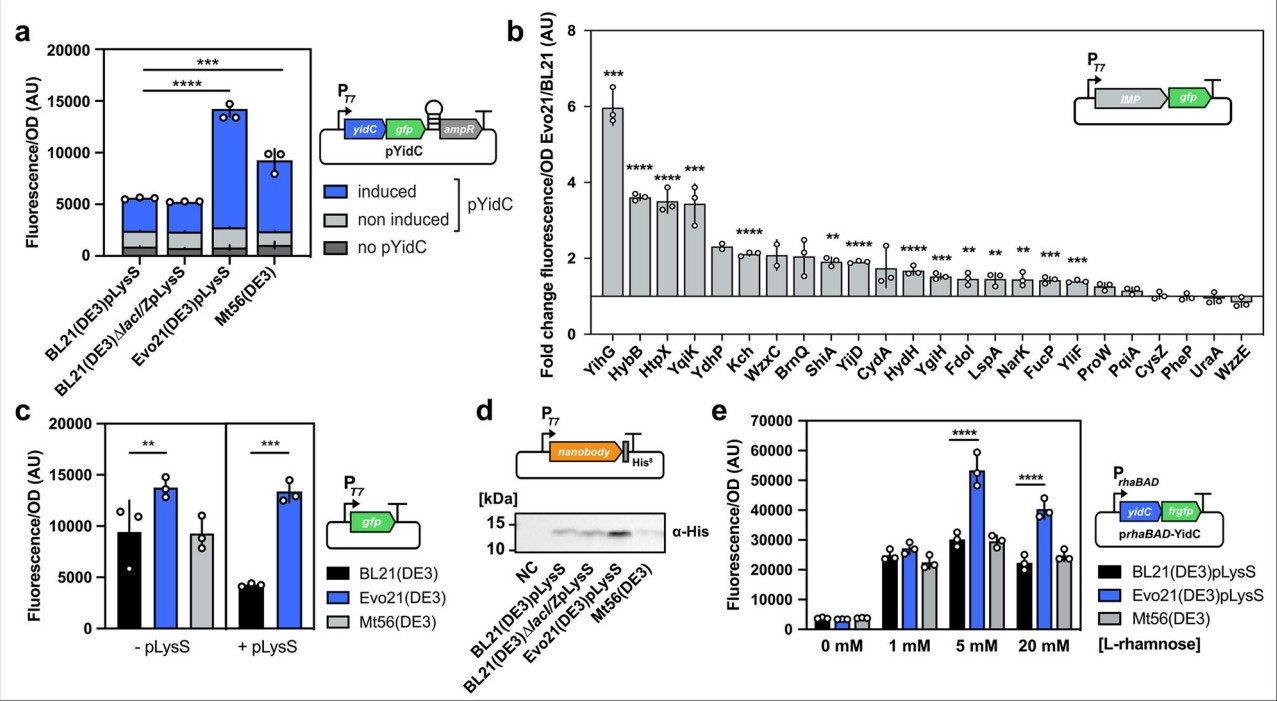

**Fig. 2 Characterisation of mutant Evo21(DE3) for toxic and non-toxic protein production. a** Production of a toxic YidC-GFP-fusion protein in Evo21(DE3) compared to the non-evolved BL21(DE3) wild-type strain, the BL21(DE3)Δ*lacI/Z* ancestor strain, and a previously evolved BL21(DE3) derivative Mt56 (DE3). On the illustrated expression vector (pYidC), YidC-GFP production is translationally coupled to ampicillin resistance. **b** Fluorescence fold change between Evo21(DE3) and BL21(DE3), producing a library of 24 proteins of the *E. coli* inner membrane proteome C-terminally fused to GFP. **c** *gfp* expression levels in Evo21(DE3) and control strains with and without co-expression of the helper plasmid pLysS. **d** Western blot showing the expression of a camelid-derived single-chain antibody (nanobody) in Evo21(DE3) and control strains. Samples were normalised to cell density before loading. **e** Production of YidC under the control of a PrhaBAD promoter allowing titration of expression L-rhamnose. All fluorescence values displayed are normalised to OD at 600 nm. Error bars indicated represent the average squared deviation from the mean (SD) of three biologically independent samples (n = 3).

expression vector, transformed it into Evo21(DE3), and expressed the construct by inducing with different rhamnose concentrations in liquid culture. With concentrations of 5 and 20 mM L-rhamnose, Evo21(DE3) produced significantly ($P \leq 0.0001$) more protein than BL21(DE3) and Mt56(DE3) (Fig. 2e).

In summary, this initial characterisation shows that the evolved strain can produce a higher titre of a range of different proteins using a T7-system-independent mechanism.

### Whole-genome sequencing and identification of a single causative mutation in Evo21(DE3).

The phenotype of Evo21(DE3) prompted us to sequence the strain using Illumina whole-genome sequencing. Two point mutations—one in the *argE* and one in the *fecB* locus (Fig. 3a)—and an insertion of a mobile IS1 element into the *rne* gene were identified. Upon reintroduction of the *argE* or *fecB* point mutations into BL21(DE3) by oligonucleotide-based recombineering, the YidC-GFP overexpression phenotype was not obtained (data not shown), whereas, when reintroducing the truncation of the *rne* locus into BL21(DE3) and the Evo21(DE3) parental BL21(DE3) Δ*lacI*Δ*lacZ* strain, the YidC-GFP overexpression tolerance phenotype was nearly identical to Evo21 (DE3) (Fig. 3b). This makes it highly likely that the *rne* IS1 insertion is the main causative mutation in Evo21(DE3).

The identified IS1 insertion causes a truncation of the encoded 1061-residue *E. coli* endoribonuclease RNase E after amino acid 702 and, therefore, a polypeptide lacking the last 359 residues of its C-terminus in Evo21(DE3) (Fig. 3c). RNase E is an essential membrane-associated enzyme involved in the maturation of both ribosomal RNA and tRNA, as well as total mRNA decay, and mediates the assembly of a multi-enzyme complex referred to as

the RNA degradosome (Fig. 3c). It has previously been shown that only the N-terminal half (1–529) of RNase E, accommodating the active catalytic domain, is essential for cell growth, and the C-terminal non-catalytic region is mostly disordered and known to function as a scaffold mediating the association of the enzymes PNPase, Rhlb and enolase[26–28].

Interestingly, a similar truncation of the *rne* locus, *rne131*, resulting in an RNase E polypeptide lacking its non-catalytic region (amino acid residues 1–584, Fig. 3c) was isolated in a screen for suppressors of a temperature-sensitive allele of the *mukB* gene[26]. A later study showed that in strains such as BL21 (DE3), introducing the *rne131* truncation caused a bulk stabilisation of mRNA degradation, including mRNA produced by T7 RNAP[29]. The *rne131* truncation was engineered into the commercially available BL21Star(DE3) with the rationale that stabilising bulk mRNA would result in increased protein production. However, following the same rationale, the commercial strain also comes with a note suggesting that it might be unsuitable for overexpression of toxic genes.

We compared the expression of six different genes that we previously found expressed better in Evo21(DE3) than in BL21 (DE3) with expression in BL21Star(DE3) and found expression levels to be highly similar between BL21Star(DE3) and Evo21 (DE3) (Fig. 4a). This provides an independent confirmation that the phenotype of Evo21(DE3) is caused by the truncation of *rne*.

### A spontaneously occurring dominant mutation in *rne* increases protein production when supplied on an auxiliary plasmid.

The way Evo21(DE3) was isolated from papillae outgrowing colonies on week-old agar plates, and because dominant *rne*

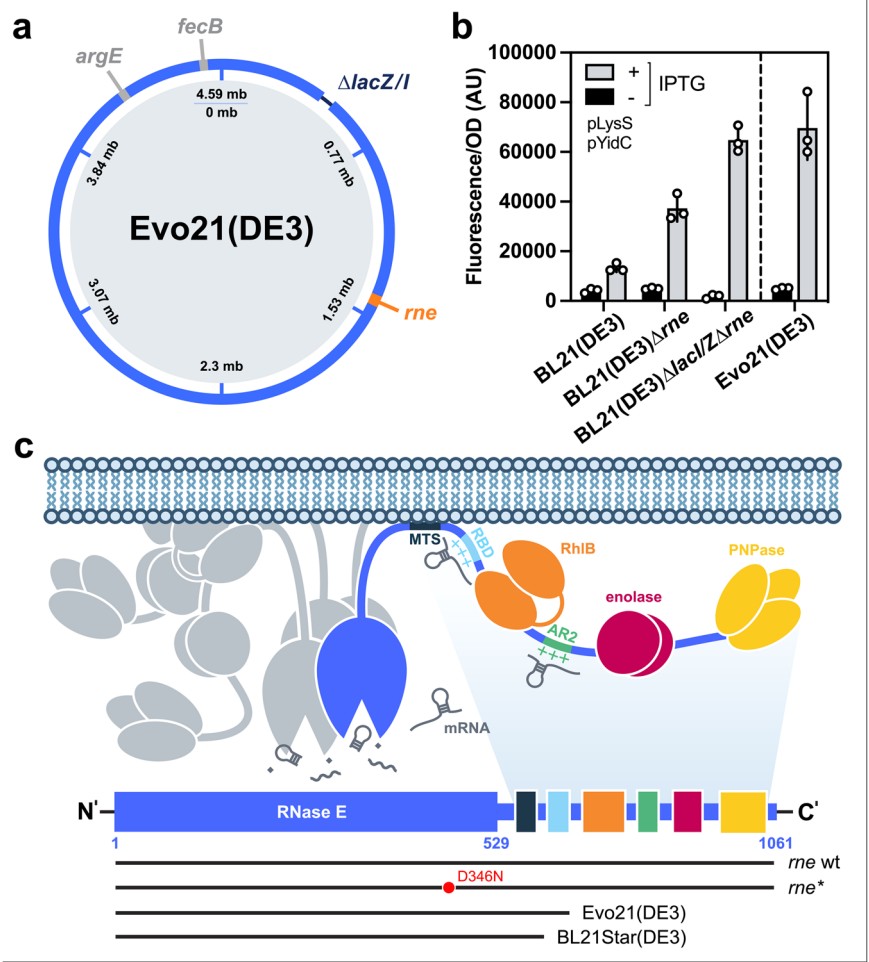

**Fig. 3 Identification of an RNase E mutation in Evo(DE3) causative for tolerance towards toxic protein production. a** Illustration of mutations in the Evo21(DE3) genome compared to the ancestor BL21(DE3). Whole-genome sequencing of mutant strain Evo21(DE3) revealed two point mutations (grey) and a truncation of the *rne* gene caused by the transposition of a mobile element IS1 into the locus. The deletion of the genomically shared homology sequence in the BL21(DE3) ancestor strain with the λDE3 area (335,401–337,123) is annotated. **b** Production levels of the toxic YidC-GFP-fusion protein in Evo21(DE3) compared to BL21(DE3), as well as BL21(DE3) and the non-evolved ancestor strain BL21(DE3)Δ*lacI*Δ*lacZ* after reintroducing the *rne* truncation by recombineering. Error bars indicated represent the average squared deviation from the mean (SD) of three biologically independent samples ($n = 3$). **c** Illustration of the *E. coli* RNA degradosome. N- and C-terminal domain of the membrane-bound essential endonuclease RNase E (blue) and the localisation of associated enzymes PNPase, Rhlb and enolase along the C-terminal non-catalytic scaffolding region are displayed. Mutations of the *rne* gene in Evo21(DE3), BL21Star(DE3) and *rne** gene harboured on pLysS-Max are indicated.

mutants previously have been observed[30], made us speculate that different *rne* variants could be studied by simple co-expression from a plasmid in the presence of wild-type *rne* on the genome. To test this idea and to compare different variants of the *rne* gene at different expression levels, we cloned *rne* variants in front of the *rhaBAD* promoter on the pLysS plasmid backbone: full-length *rne*, the Evo21(DE3) and BL21Star(DE3) truncated versions and a version with a further truncation in the membrane-binding domain of RNase E. However, none of these constructs showed any positive effect on YidC-GFP expression (Supplementary Fig. 3).

Serendipitously, we isolated a spontaneously occurred *rne* mutant, called *rne**, and included it in our analysis. We found that *rne** provided on the pLysS plasmid (hereafter called pLysS-Max) could increase YidC-GFP production even further than Evo21(DE3) (Fig. 4b). The *rne** mutation converts the essential[31] aspartate residue in position 346 to an asparagine in the so-called DNase I subdomain of RNase E involved in chelating an essential $Mg^{2+}$ ion. The aspartate residue is believed to act as a general base to activate the attacking water essential for the catalytic

activity of the enzyme[32]. The replacement of Asp-346 with the polar amino acid Asn was previously shown to decrease RNA cleavage by about 25-fold[32]. The effect of expressing *rne** was not gene-specific as the effect was preserved for three out of four other tested genes (Fig. 4b). This provides an alternative demonstration that manipulating with *rne* severely affects recombinant protein production and provides a simple tool, in the form of an auxiliary pLysS-Max plasmid that can be transformed into other strains, to improve protein production titre.

**Utilising RNase E autoregulation to build a biosensor of RNase activity in the cell.** The positive effect of *rne* truncations such as *rne131* on protein production was previously assumed to be due to the stabilisation of the recombinant mRNA[29]. However, the observation that the *rne* truncation in Evo21(DE3) leads to tolerance of toxic gene expression suggests a broader role involving balancing of RNA levels more globally.

Autoregulation allows RNase E to continuously adjust its synthesis to that of its substrates by controlling the degradation

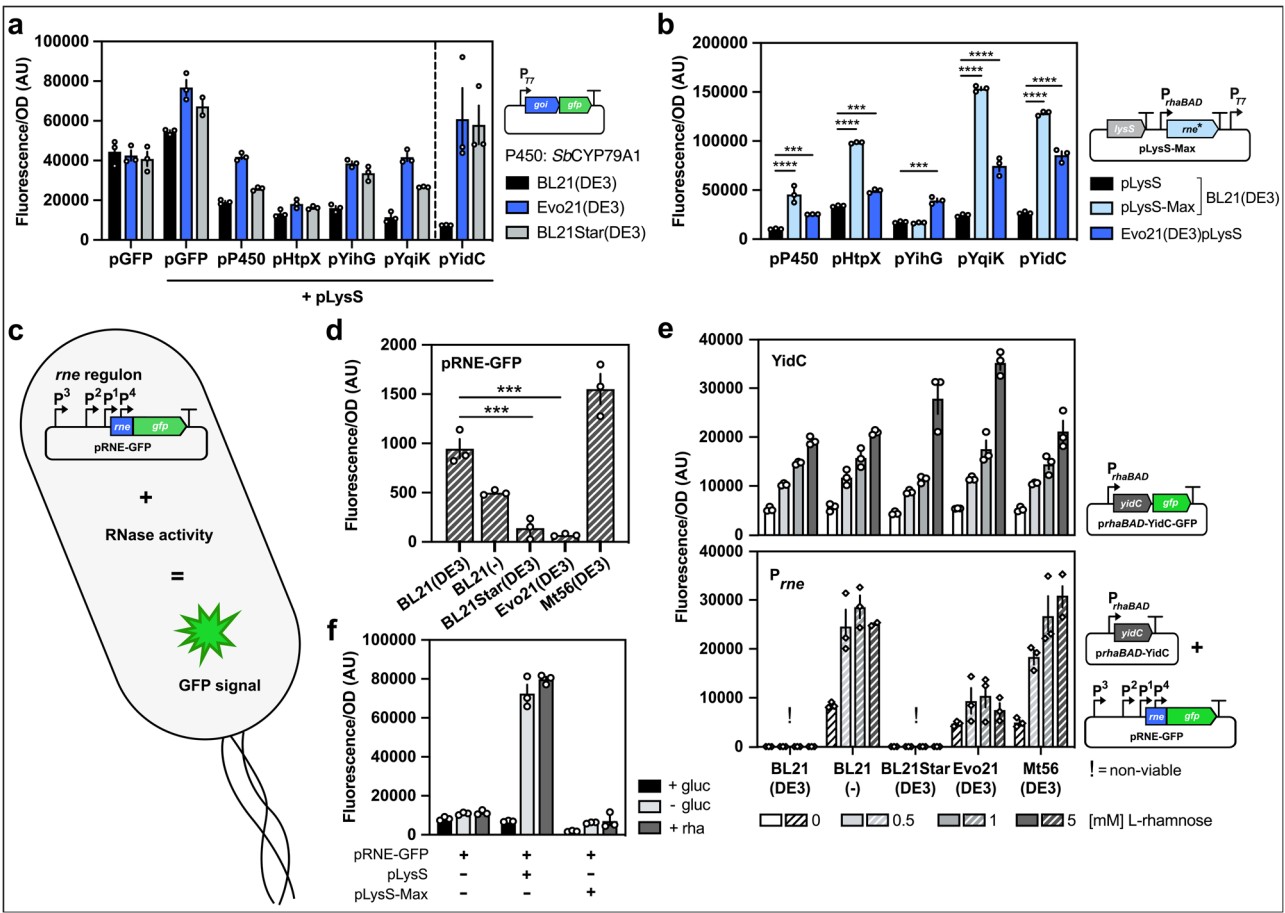

**Fig. 4 Effect of *rne* mutations on protein production and RNase activity levels in *E. coli*. a** Heterologous production of a variety of non-toxic and toxic GFP-fusion proteins in Evo21(DE3) and BL21Star(DE3), both harbouring different truncations of the *rne* gene, compared to BL21(DE3). **b** Expression of the same genes in Evo21(DE3) is compared to expression in BL21(DE3) when co-expressing either the auxiliary plasmid pLysS or pLysS-Max. **c** Schematic illustration of the plasmid pRNE-GFP designed to report on RNase activity. Promoters controlling *rne* expression are indicated (P1–4)[49,50]. The expression level of the *rne* gene can be monitored in vivo via GFP fluorescence signal. **d** Exploration of the pRNE-GFP reporter plasmid in BL21(DE3) derivatives. Different levels of RNase activity can be monitored in the strains. **e** Titration of *yidC* expression in Evo21(DE3), BL21Star(DE3) and BL21(DE3) via increasing levels of L-rhamnose and its effect on the *rne* regulon expression reporting RNase activation in the cell during toxic membrane protein production. Upper half: Fluorescence corresponds to YidC-GFP production levels. Lower half: Cells harbour both the *rne* reporter plasmid (pRNE-GFP) and pP*rhaBAD*-YidC controlling *yidC* expression (no GFP fusion). Fluorescence levels correspond to GFP produced under the control of the *rne* regulon. **f** Effect of the pLyS-Max auxiliary plasmid on RNase activity. Plasmids pLysS and pLysS-Max are co-expressed along with pRNE-GFP in BL21(DE3) cells. To repress leaky expression of the rhamnose promoter controlling *rne** expression on pLysS-Max, 0.4% glucose was added where indicated. Error bars indicated represent the average squared deviation from the mean (SD) of three biologically independent samples (*n* = 3).

rate of its own mRNA[33,34]. This could work as a biosensor for RNase E activity by fusing the promoter and 5′ end of *rne* with a genetic reporter, as previously demonstrated with *lacZ*[34]. To explore RNase E activity in our evolved strains for recombinant protein production, we constructed a similar RNase E biosensor (pRNE-GFP) using GFP as a reporter (Fig. 4c).

We transformed the pRNE-GFP reporter into BL21, BL21(DE3), BL21Star(DE3), Evo21(DE3) and Mt56(DE3) and monitored fluorescence in a microplate reader under conditions similar to the typical protein production scenario. Surprisingly, under these conditions, fluorescence was 14-fold reduced in Evo21(DE3) and sevenfold reduced in BL21Star(DE3) compared to the ancestral BL21(DE3 (Fig. 4d). Given that Evo21(DE3) behaves almost identically to BL21Star(DE3) and that the *rne131* truncation has been shown to cause a bulk stabilisation of mRNA degradation[29], this suggests that low fluorescence from our reporter correlates with increased bulk mRNA stabilisation. Interestingly, fluorescence from the reporter was reduced approximately twofold in BL21 compared to BL21(DE3),

suggesting an effect of the λDE3 locus itself on RNase E activity in the cell.

Next, we wanted to explore if the expression of YidC-GFP affected RNase E activity in the different strains. To this end, because YidC-GFP cannot be expressed from the T7 promoter in BL21 (no T7 RNA polymerase) or BL21(DE3) (no growth), we expressed it from the *rhaBAD* promoter construct (Fig. 2e) using different concentrations of rhamnose. This showed that YidC-GFP levels could be titrated with rhamnose and confirmed higher expression in BL21Star(DE3) and Evo21(DE3) than in the other strains at high rhamnose concentrations (Fig. 4e, upper half).

To monitor fluorescence from the RNase E GFP reporter, we then deleted GFP from the YidC construct and attempted to co-transform it with pRNE-GFP into different strain backgrounds (Fig. 4e, lower half). However, we were unable to recover and grow transformants in BL21(DE3) and BL21Star(DE3), suggesting a lethal imbalance in RNA levels caused by the presence of these two plasmids. We were able to recover and grow double transformants in BL21, Evo21(DE3), and Mt56(DE3) and

monitor fluorescence as an indication of RNase E activity. Fluorescence increased to high levels upon increasing the concentration of rhamnose in BL21 and Mt56(DE3), but fluorescence levels were at least 2.5-fold lower (at 1 mM rhamnose) in Evo21(DE3) and hardly increased upon further rhamnose addition. This suggests that RNase E activity towards bulk mRNA is increased when *yidC* expression is increased but that the activity is lower in Evo21(DE3) than in the other strains (Fig. 4e).

Finally, we explored the effect of the pLysS-Max auxiliary plasmid (harbouring the mutant *rne\**) on RNase E activity by co-transforming it along with pRNE-GFP into BL21(DE3). As controls, we included a strain co-transformed with pLysS and pRNE-GFP and BL21(DE3) transformed only with pRNE-GFP. Because we previously observed effects on RNase E activity due to the presence of the λDE3 locus, we repressed leaky expression of T7 RNAP from the *lacUV5* promoter by adding glucose to the medium and titrated *rne\** levels with rhamnose. In the absence of glucose and the presence of pLysS, we observed an increase in fluorescence which was repressed by expression of *rne\** (Fig. 4f). This shows that the pLysS-Max plasmid can be used to regulate RNase E activity in the cell.

## Discussion

Gene toxicity is not restricted to membrane proteins, and the protein burden problem has received attention since the 1950s'[35],[36]. No uniform theory explains the protein burden, but commonly it is observed that ribosomal RNA is degraded, similar to what is observed with nutrient starvation[37]. A recent study explored the global transcriptomic response to overexpression of a set of 45 different genes in *E. coli* and found highly gene-specific activation of different host responses such as fear vs greed, metal homoeostasis, respiration, protein folding and amino acid and nucleotide biosynthesis[6]. Thus, gene overexpression shows many similarities with other types of cellular stress, but the role of RNA degradation in stress responses is not well-understood[38].

By restricting the evolutionary solution space of the protein production workhorse BL21(DE3), we have discovered a key role for RNA stability in determining the toxicity and productivity of recombinant protein production in *E. coli*. This provides an example that engineering the evolution of a microbe can help develop useful traits and discover new mechanisms—even with well-characterised systems such as T7-based expression in BL21(DE3). In contrast to previously evolved BL21(DE3)-derivatives, Evo21(DE3) does not contain mutations in T7 RNAP, and it is therefore also able to produce higher levels of non-toxic proteins such as GFP. The observation that RNA is a key player in gene expression toxicity is in line with recent work demonstrating that expression of mRNA codon variants influences the fitness of *E. coli* independently of translation[39].

The causative mutation in *rne* in the evolved Evo21(DE3) strain is similar to, but distinct from, the *rne* mutation engineered into BL21Star(DE3). This is surprising because the main mechanism assumed for BL21Star(DE3) was stabilisation of the recombinant mRNA, and it was therefore thought to be sub-optimal for toxic proteins. This belief is at least partly based on observations that T7 RNAP transcripts are unusually RNase E-sensitive[40]. However, as shown here, the positive effect of an *rne* truncation appears not to be specific to T7 RNAP transcripts, and BL21Star(DE3) previously did show great capacity to produce a number of different membrane proteins[41], and the RNase E inhibitor RraA was identified in a screen for genes whose co-expression could suppress membrane protein-induced toxicity[42]. Possibly, the stabilisation of mRNA caused by the *rne* mutations in BL21Star(DE3) and Evo21(DE3), or by overexpression of *rraA*,

is not restricted to the recombinant mRNA but also stabilises other essential mRNAs that are otherwise destabilised by resource allocation to producing the recombinant molecule[43]. It is interesting to note that RNase E encompasses a membrane-binding domain, and the gene now twice has been implicated in membrane protein production stress. Further studies should examine this possible connection.

The *rne\** mutation D346N is located in the N-terminal endoribonucleolytic domain of RNase E, in the highly conserved region of the RNase E/RNase G family called high similarity region 2 (HSR2)[44],[45]. The region contains an arginine-rich domain that may contact the RNA[45], and removal of the negative charge in position 346 negatively influences the catalytic activity of the enzyme[31],[32]. Similarly, a mutation in RNase G (G341S), only five residues away from D346 in an almost completely conserved region[44], was previously shown to be defective in the degradation of a specific mRNA but proficient in processing of 16s rRNA[46].

Using a GFP-based biosensor, we monitored RNase E activity in different BL21 variants in the presence or absence of pLysS and the YidC overexpression vector. The biosensor was based on the 5' region of *rne* fused to *gfp* and a highly similar construct was previously shown to decrease expression in response to increased RNase E activity due to destabilisation of the *rne* part of the mRNA[34]. However, RNase E substrate recognition is highly complex and different domains coordinate a hierarchy of efficiencies with which different RNA targets are attacked[31]. The decrease in fluorescence from the pRNE-GFP reporter in BL21Star(DE3) shown here suggests an increase in RNase E activity towards the reporter mRNA, but in the same strain bulk mRNA was previously shown to be stabilised[29]. Furthermore, the work by Lopez et al. indicated no increase in expression of the *rne131* truncated mutant compared to the wild-type *rne*, which would have been expected if the mutant *rne* affected autoregulation. Regardless of the underlying mechanism, low fluorescence from the pRNE-GFP reporter was observed here in combination with the truncated versions of *rne* in Evo21(DE3) and BL21Star(DE3), and upon co-expression of the *rne\** mutant, and higher fluorescence was observed in response to both the presence of the T7 RNA polymerase in itself and when increasing the expression of a foreign gene like *yidC*. This supports a central role for RNase E activity in determining recombinant protein production efficiency. The pRNE-GFP RNase biosensor may come in handy as a tool for diagnosing RNase activity in other settings and could facilitate a broader understanding of the role of RNA stability in cell factory engineering.

## Methods

**Bacterial strains and plasmids.** *E. coli* production strain BL21(DE3)[47] (Invitrogen, Waltham, MA, USA; Catalogue number: C600003) was used to generate BL21(DE3) Δ*lacI*Δ*lacZ* (described below) employed here to isolate derivate Evo21(DE3). BL21 (New England Biolabs, Ipswich, MA, USA; Catalogue number: C2530H), BL21(DE3) and Mt56(DE3)[12] were used as control strains for experiments benchmarking Evo21 (DE3)'s performance. Plasmid pLysS was utilised to limit basal T7 RNAP expression via co-expression of the natural T7 RNAP inhibitor T7 lysozyme that binds the polymerase upon expression and inactivates it.[23] All proteins produced in this study are listed in Supplementary Table 1. Membrane proteins were produced as C-terminal GFP-His$_8$ fusion proteins from a pET28+-derived expression vector[48]. For the creation of expression vector pET28-*yidC-gfp-hp-bla* (pYidC), used for the evolution experiment which led to the isolation of Evo21(DE3), a 25 bp hairpin loop sequence (5'-TGA-CATAGGAGGTCCTCCTATGTCA-3') for strong translational coupling was inserted downstream of the His-tagged *gfp* sequence followed by an in-frame ß-lactamase gene (*bla*) conferring resistance to the antibiotic ampicillin as described in Rennig et al.[19]. All expression vectors encode a *lacI* gene under the control of its native *lacI* promoter to reduce T7 RNAP production prior to IPTG induction in λDE3 strains. Plasmid pLysS was chosen to function as the backbone for co-expression of different *rne* variants to avoid the burden of having three plasmids present simultaneously in the system. All genes expressed on the pLysS backbone were cloned seamlessly in between *lysS* terminator and T7Φ3.8 promoter controlling *T7lysS* expression flanked by additional terminator sequences: T7 terminator (upstream) and *rrnc* terminator (downstream).

RNase E reporter plasmid pRNE-GFP was constructed similar to the previously described vector pEZIO1[34]: The complete *rne* 5' UTR (555 bp) comprising promoters P1 to P4[49,50] and the first 181 codons of the *E. coli rne* gene were joined in-frame to the second codon of a *gfp* gene. Cells were grown aerobically at either 37 or 30 °C, and 200 rpm in lysogeny broth (LB) (Difco) supplemented with 50 μg/ml kanamycin, 25 μg/ml chloramphenicol or 100 μg/ml ampicillin depending on the resistance marker of the plasmid used.

**Genome engineering.** Parental strain BL21(DE3)*ΔlacIΔlacZ* was created using λRED recombineering. BL21(DE3) was transformed with plasmid pSIM19[51], harbouring genes for λRED recombination proteins *exo*, *bet* and *gam* under the control of a heat-inducible promoter. Recombineering was performed using a *tetA* integration cassette flanked by 50-bp long homology sequences to the λDE3 homology area in *E. coli* BL21(DE3). The *tetA* integration cassette was subsequently removed via the counter selection on NiCl₂, establishing a total deletion of 1,722 bp in genome position 335,401–337,123 verified via sanger sequencing. All mutations identified in Evo21(DE3) via next-generation sequencing were re-engineered using MAGE[52] and oligos 1–9 (Supplementary Table 2).

**Isolation of Evo21(DE3).** To evolve a new BL21(DE3) derivate with enhanced tolerance towards toxic protein expression, BL21(DE3)*ΔlacIΔlacZ* was transformed with the expression vector pYidC (described above) expressing *yidC* translationally coupled to expression of a ß-lactamase encoding gene facilitating ampicillin resistance, alongside with plasmid pLysS. Successful transformants were inoculated into LB medium supplemented with 50 μg/ml kanamycin and 25 μg/mL chloramphenicol and grown overnight at 37 °C and 200 rpm in 5-mL falcon tubes. Production of membrane protein YidC C-terminally fused to GFP allows identification of BL21(DE3)*ΔlacIΔlacZ*-derived mutants efficiently producing YidC in the cytoplasmic membrane upon illumination with UV light. Translational coupling to ampicillin resistance provides an additional layer of selection. Only cells producing the membrane protein fusion protein can grow on a growth medium supplied with the antibiotic. This limits the evolutionary space of arising mutations to ones without a negative impact on production levels.

To allow colony formation prior to membrane protein production, 100 μL of undiluted overnight culture had been plated on layered agar plates. In total, 25 mL LB-agar, containing both 50 μg/ml kanamycin and 25 μg/mL chloramphenicol, was poured on top of a hardened 5 mL LB-agar layer, additionally supplemented with 150 μg/mL ampicillin and 1 mM IPTG. Slow diffusion of IPTG and ampicillin into the upper layer gives cells plated on top sufficient time to form colonies before toxic membrane protein production is induced. Plates were incubated at 37 °C for several days and monitored daily for mutant papillae outgrowing the population. Multiple fluorescent colonies could be isolated during the first week, showing higher fluorescence levels than BL21 (DE3) in an initial plate reader experiment screening for GFP fluorescence. One isolate, subsequently named Evo21(DE3), consistently showed high GFP levels and was chosen for further studies. To cure Evo21(DE3) of the *yidC-gfp* expression vector used during the evolution experiment, the CRISPR-Cas9 based pFREE plasmid curing system was employed[22]. For all experiments performed thereafter, Evo21(DE3) was always freshly transformed with different vectors.

**Plate reader experiments.** For growth assays, strains were cultured overnight in 5 mL LB liquid growth medium. Dilutions of 1:50 were grown aerobically for 24 h in 96-well plates at 37 °C and 200 rpm using Gas Permeable Adhesive Seal (Thermo Fisher Scientific, Waltham, MA, USA) to avoid evaporation. In all, 1 mM IPTG was added at $OD_{600} = 0.3$. Growth (absorbance at 600 nm) and fluorescence (GFP: excitation at 485 nm, emission at 528 nm) was measured in 20 min intervals while continuous shaking using a Synergy H1 plate reader (BioTek Instruments, Winooski, VT, USA). All measurements were performed in triplicate.

**Western blots.** Whole-cell lysate analysis via SDS-PAGE and immunoblotting[53] 1 ODU of cells was lysed using CelLytic B 2× Cell lysis reagent (Sigma-Aldrich, Saint Louis, MO, USA) and run on standard SDS-PAGE using 12% polyacrylamide gels with 5x reducing sample buffer (8 M Urea, 10% SDS). In total, 15 μL of each sample were loaded and run for 30 min at 200 V. SDS-PAGEs were blotted on nitrocellulose membranes using the iBlot™ 1 gel transfer system (Invitrogen, Waltham, MA, USA), and His-tagged nanobody was detected using α-His mouse antibody (05-949, LOT: 3033896, 1:1000, Merck Millipore, Darmstadt, Germany) diluted in 2% skim milk, followed by incubation with a secondary HRP-conjugated rabbit α-mouse IgG (AP160P, 1:10,000, Merck Millipore, Darmstadt, Germany) diluted in TBS-T. Proteins were visualised using the Amersham ECL Prime Western Blotting System (GE Healthcare, Chicago, IL, USA) according to the manufacturer's instructions and an Amersham gel imaging system.

**Next-generation sequencing.** DNA purification was performed using DNeasy Blood & Tissue Kit 50 (Qiagen, Hilden, Germany) starting with 1 mL overnight culture according to the manufacturer's instructions. Isolated DNA was eluted twice using 200 μL 10 mM Tris (pH = 8.5). The genomic libraries were generated using the TruSeq DNA HT Library Prep Kit (Illumina, San Diego, CA, USA). Whole-genome sequencing

was performed using the Illumina NextSeq 500 system and sequencing adapters D701 (ATTACTCG) and D501 (TATAGCCT). Data analysis was performed using the breseq software developed by D. Deatherage and J. Barrick[54].

**Statistics and reproducibility.** All experiments were performed in biological triplicate. Error bars and significance values were calculated using the program PRISM. Error bars indicated represent the average squared deviation from the mean (SD). A one-way ANOVA with Dunnett's multiple comparison test was employed to evaluate differences in expression levels of recombinant protein between Evo21(DE3) and other expression hosts. $P$ values <0.05 were accepted statistically significant. The different significance levels indicated as stars in figures correspond to *$P$ value <0.05, **$P < 0.01$, ***$P < 0.001$ and ****$P < 0.0001$.

**Reporting summary.** Further information on research design is available in the Nature Research Reporting Summary linked to this article.

## Data availability
All whole-genome sequencing data connected to this study has been deposited to the Sequence Read Archive (SRA) and can be accessed via BioProject ID PRJNA741995. All other data will be made available upon reasonable request to the corresponding author. The uncropped and unedited western blot image corresponding to Fig. 2d is included as Supplementary Fig. 4.

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

## Acknowledgements
We thank Prof. Daniel O. Daley for sharing the membrane protein expression vector library. This work was supported by a grant from the Novo Nordisk Foundation NNF17OC0027752 to M.H.H.N.

## Author contributions
S.A.H.H. performed the experiments that led to the creation of this article and analysed and interpreted the data. M.H.H.N. and S.A.H.H. contributed to shaping the idea of the manuscript and the experimental design process and wrote the manuscript. All authors revised, read and approved the final manuscript.

## Competing interests
The authors declare the following competing interests. A patent application has been filed by the Technical University of Denmark with the inventors being M.H.H.N. and S.A.H.H. under European Patent Application No. 21177466.6. The application covers the plasmid pLysS-Max described within this manuscript.
