## [Peer Review File · Communications Biology]

Reviewers' Comments:

Reviewer #1:

Remarks to the Author:

The manuscript describes the mutation in the *rne* gene responsible for the reduced toxicity from membrane protein overexpression. This finding was previously reported, however, they found another mutation in the *rne* gene which better improved toxic protein production. Followings are points authors need to address/consider to improve the manuscript.

[Major comments]

1) In addition to mutation in the T7 RNAP gene, mutation in the *lacI* gene is responsible for the reduced toxicity from membrane protein overexpression. [Scientific Reports. 5, 16076.(2015); ACS Synthetic Biology 6(9): 1766-1773 (2017)]. Please describe and discuss this.

2) The GFP-based system has yielded great results and provided the proof of concept. However, GFP is a soluble and stable protein. For the tested proteins, will GFP tag change their functions, stabilities or the associated cellular toxicity?

[Minor comments]

1) There are two *lacI* genes in the BL21(DE3) chromosome. Were both deleted the *lacI* deletion mutant?

2) P6 "we Sanger sequenced the T7 RNAP gene, which confirmed an absence of mutations." -> How about the *lacI* gene?

3) P10 "We found it challenging to construct these plasmids as an unusual high number were isolated with spontaneously occurring mutations in *rne*." -> I don't understand this. Please rephrase this.

4) Typos are
P2L7: 200.000
P2L8: 128.000
P6 L10: 2,1

5) "Evo21" should be changed to "Evo21(DE3)": P8L32, P10L11, P14L1

Reviewer #2:

Remarks to the Author:

This is a very interesting story about the role of RNase E in the toxicity mediated by overexpression of heterologous proteins in derivatives of the BL21(DE3) strain of *E. coli*. Several previous studies demonstrated that toxicity can be mitigated by mutations in the T7 polymerase promoter. Here, the authors used a clever experimental design to prevent T7 promoter mutations and facilitate the isolation of alternative mutations with a similar phenotype. They show convincingly that reduction of RNase E activity is responsible for reduced toxicity and increased protein production. They then recapitulate the results for several different membrane proteins, and they isolate a dominant mutant of RNase E with a similar phenotype. These results show an unexpected connection between global RNA metabolism and the efficiency of heterologous protein production in *E. coli*.

Specific comments:

p11, lines 1-12: The interpretation of RNase reporter experiments is unclear. If the reporter RNA is degraded by RNase E, this should lead to increased expression of the reporter in the Evo21(DE3) strain. If RNase E is autoregulated at the transcriptional level, this should again lead to increased expression of the reporter in the RNase knockout strain. However, figure 4 shows the opposite

result: a decreased expression of the reporter in Evo21(DE3). A more thorough discussion of the mode of action of the reporter would help interpret subsequent experiments with this reporter.

A recent paper by Mittal et al (2018) described toxicity mediated by heterologous expression of RNA in *E. coli*, in the absence of translation. Could the findings described here provide a clue as to the mechanisms of RNA toxicity?

p2, line 18: MPs are involved in (...) creation of energy -> please rephrase, creation of energy is not possible.

p6, line 31: The finding that heterologous protein production in Evo21(DE3) is higher than in the parental strain does not, by itself, prove that T7 polymerase is not affected in Evo21(DE3). For example in Fig 2, Mt56(DE3) shows higher expression of heterologous proteins than BL21(DE3), despite harboring a mutation in T7 polymerase.

Reviewer #3:

Remarks to the Author:

Gene expression toxicity is a major bottleneck in biotechnology especially in heterologous expression systems. In this manuscript, the author restrict the solution space for *E. coli* BL21 to evolve tolerance and isolate a mutant strain *E. coli* Evo21(DE3) with a truncation in the essential RNase E. This suggests that RNA stability may play an important role in gene expression toxicity. The evolved *rne* truncation is similar to a mutation previously engineered into the commercially available BL21Star(DE3), this may challenge the existing assumption that this strain is unsuitable for expression of toxic proteins. Another dominant mutation in a presumed substrate binding site of RNase E was obtained, which improves protein production further when provided as an auxiliary plasmid. The results are helpful for the problem of gene expression toxicity.

In the manuscript (Fig.2, b), the author tested fluorescence fold change between Evo21(DE3) and BL21(DE3) with a library of 24 proteins of the *E. coli* inner membrane proteome C-terminally fused to GFP. More testing of non-membrane proteins is of great significance for the application of host bacteria in metabolic engineering.

RNase E played a key role in RNA degradation in the cyanobacterium *Anabaena* PCC 7120 (Nucleic Acids Research , doi:10.1093/nar/gkaa092), whether this function compatible with its function in gene expression toxicity?

There are a lot of non-standard spelling in the manuscript, including the name of strain, the name of protein and the Latin in the reference et al, which will affect the credibility of the data.

Point-to-point response to Reviewers' comments

*Responses to specific comments are indicated with an **R** and in **bold** below. A revised version of the manuscript with indications of changes (track-changes) has been submitted together with a clean new version of the manuscript. Page and line numbers indicated below correspond to the manuscript version with changes indicated.*

Reviewer #1 (Remarks to the Author):

The manuscript describes the mutation in the rne gene responsible for the reduced toxicity from membrane protein overexpression. This finding was previously reported, however, they found another mutation in the rne gene which better improved toxic protein production. Followings are points authors need to address/consider to improve the manuscript.

[Major comments]

1) In addition to mutation in the T7 RNAP gene, mutation in the lacI gene is responsible for the reduced toxicity from membrane protein overexpression. [Scientific Reports. 5, 16076.(2015); ACS Synthetic Biology 6(9): 1766-1773 (2017)]. Please describe and discuss this.

R: Thank you for pointing this out. We added the information and reference on page 3, lines 10-11.

2) The GFP-based system has yielded great results and provided the proof of concept. However, GFP is a soluble and stable protein. For the tested proteins, will GFP tag change their functions, stabilities or the associated cellular toxicity?

R: Yes, this is certainly possible. But as we also point out and demonstrate: (1) The YidC-GFP construct is toxic, and (2) We observe increased expression in Evo21(DE3) also with e.g., a nanobody that is not tagged with GFP

[Minor comments]

1) There are two lacI genes in the BL21(DE3) chromosome. Were both deleted the lacI deletion mutant?

R: No, we only deleted lacI from the native locus. This should be clear from the description on page 4, lines 8-9 and page 17, lines 1-2.

2) P6 "we Sanger sequenced the T7 RNAP gene, which confirmed an absence of mutations." -> How about the lacI gene?

R: The described Sanger sequencing was part of the initial screening designed only to avoid well-known mutations e.g. in T7 RNAPol. As described, we later whole-genome sequenced Evo21(DE3) and found no mutations in *lacI*.

3) P10 "We found it challenging to construct these plasmids as an unusual high number were isolated with spontaneously occurring mutations in *rne*." -> I don't understand this. Please rephrase this.

R: Thank you for pointing this out. This information is of minor importance since we anyway succeeded in assembling the constructs. We removed the sentence.

4) Typos are
P2L7: 200.000
P2L8: 128.000
P6 L10: 2,1

R: Thank you! The typos were corrected

5) "Evo21" should be changed to "Evo21(DE3)": P8L32, P10L11, P14L1

R: Thank you! The names were corrected

Reviewer #2 (Remarks to the Author):

This is a very interesting story about the role of RNase E in the toxicity mediated by overexpression of heterologous proteins in derivatives of the BL21(DE3) strain of *E. coli*. Several previous studies demonstrated that toxicity can be mitigated by mutations in the T7 polymerase promoter. Here, the authors used a clever experimental design to prevent T7 promoter mutations and facilitate the isolation of alternative mutations with a similar phenotype. They show convincingly that reduction of RNase E activity is responsible for reduced toxicity and increased protein production. They then recapitulate the results for several different membrane proteins, and they isolate a dominant mutant of RNase E with a similar phenotype. These results show an unexpected connection between global RNA metabolism and the efficiency of heterologous protein production in *E. coli*.

Specific comments:

p11, lines 1-12: The interpretation of RNase reporter experiments is unclear. If the reporter RNA is degraded by RNase E, this should lead to increased expression of the reporter in the Evo21(DE3) strain. If RNase E is autoregulated at the transcriptional level, this should again lead to increased expression of the reporter in the RNase knockout strain. However, figure 4 shows the opposite result: a decreased expression of the reporter in Evo21(DE3). A more thorough discussion of the mode of action of the

reporter would help interpret subsequent experiments with this reporter.

R: Thank you very much for pointing this out! We agree that the description of these results was confusing, which relates to the complexity of the different functions of RNase E (in terms of activities towards different RNA species such as rRNA, tRNA, and mRNA, and autoregulation). We rephrased the results and discussion sections to more correctly describe our observations and how they might relate to what is known about RNase E. See page 11, lines 7-10 and page 14, lines 17-31.

A recent paper by Mittal et al (2018) described toxicity mediated by heterologous expression of RNA in E. coli, in the absence of translation. Could the findings described here provide a clue as to the mechanisms of RNA toxicity?

R: Thanks this is a great suggestion and reference – we added a sentence in the discussion to indicate this connection – page 13, lines 25-27

p2, line 18: MPs are involved in (...) creation of energy -> please rephrase, creation of energy is not possible.

R: Thank you for pointing this out! We changed it to “energy conversion”

p6, line 31: The finding that heterologous protein production in Evo21(DE3) is higher than in the parental strain does not, by itself, prove that T7 polymerase is not affected in Evo21(DE3). For example in Fig 2, Mt56(DE3) shows higher expression of heterologous proteins than BL21(DE3), despite harboring a mutation in T7 polymerase.

R: Thank you for pointing this out! We agree that it is not definite proof, but the fact that the soluble, non-toxic proteins GFP and the Nanobody are expressed at higher levels could indicate that T7 RNAPol activity is not compromised (the data referred to in Fig. 2 is on a toxic protein – this is probably why Mt56 is better than BL21). We did a minor adjustment of the sentence to indicate this. Page 6, lines 31.

Reviewer #3 (Remarks to the Author):

Gene expression toxicity is a major bottleneck in biotechnology especially in heterologous expression systems. In this manuscript, the author restrict the solution space for E. coli BL21 to evolve tolerance and isolate a mutant strain E. coli Evo21(DE3) with a truncation in the essential RNase E. This suggests that RNA stability may play an important role in gene expression toxicity. The evolved rne truncation is similar to a mutation previously engineered into the commercially available BL21Star(DE3), this may challenge the existing assumption that this strain is unsuitable for expression of toxic

proteins. Another dominant mutation in a presumed substrate binding site of RNase E was obtained, which improves protein production further when provided as an auxiliary plasmid. The results are helpful for the problem of gene expression toxicity. In the manuscript (Fig.2, b), the author tested fluorescence fold change between Evo21(DE3) and BL21(DE3) with a library of 24 proteins of the E. coli inner membrane proteome C-terminally fused to GFP. More testing of non-membrane proteins is of great significance for the application of host bacteria in metabolic engineering.

R: Thank you for this suggestion. We certainly agree that it would be interesting to test more proteins. However, in figures 2c and 2d we demonstrate increased production of the soluble proteins GFP and Nanobody and we have also showed this for a number of P450 enzymes. Surely this is sufficient to support the overall conclusions in the manuscript?

RNase E played play a key role in RNA degradation in the cyanobacterium *Anabaena* PCC 7120 (Nucleic Acids Research , doi:10.1093/nar/gkaa092), whether this function compatible with its function in gene expression toxicity?

R: We are not entirely sure we understand this comment or the relevance of the reference. Surely, RNase E could serve similar roles in other organisms, but we don't have any data to support this.

There are a lot of non-standard spelling in the manuscript, including the name of strain, the name of protein and the Latin in the reference et al, which will affect the credibility of the data.

R: Thank you for pointing this out. We did a thorough spell-check of the document and found some mistakes, in particular related to US/British English. To our knowledge et al. is used correctly in the manuscript, but we hope the editorial team can assist with the final polishing of this.

Reviewers' Comments:

Reviewer #1:

None

Reviewer #3:

Remarks to the Author:

I accepted the author's response and recommend the article for publication.